# Adaptive Structure Induction for Aspect-based Sentiment Analysis with Spectral Perspective

**Hao Niu, Yun Xiong,* Xiaosu Wang, Wenjing Yu, Yao Zhang,**
**Zhonglei Guo**
Shanghai Key Laboratory of Data Science, School of Computer Science, Fudan University
{hniu18, yunx, xswang19, yaozhang, guozl18}@fudan.edu.cn
{wjyu21}@m.fudan.edu.cn

## Abstract

Recently, incorporating structure information (e.g. dependency syntactic tree) can enhance the performance of aspect-based sentiment analysis (ABSA). However, this structure information is obtained from off-the-shelf parsers, which is often sub-optimal and cumbersome. Thus, automatically learning adaptive structures is conducive to solving this problem. In this work, we concentrate on structure induction from pre-trained language models (PLMs) and throw the structure induction into a spectrum perspective to explore the impact of scale information in language representation on structure induction ability. Concretely, the main architecture of our model is composed of commonly used PLMs (e.g., RoBERTa, etc.), and a simple yet effective graph structure learning (GSL) module (graph learner + GNNs). Subsequently, we plug in Frequency Filters with different bands after the PLMs to produce filtered language representations and feed them into the GSL module to induce latent structures. We conduct extensive experiments on three public benchmarks for ABSA. The results and further analyses demonstrate that introducing this spectral approach can shorten Aspects-sentiment Distance (AsD) and be beneficial to structure induction. Even based on such a simple framework, the effects on three datasets can reach SOTA (state-of-the-art) or near SOTA performance. Additionally, our exploration also has the potential to be generalized to other tasks or to bring inspiration to other similar domains. [1]

## 1 Introduction

Aspect-based sentiment analysis (ABSA) is designed to do fine-grained sentiment analysis for different aspects of a given sentence (Vo and Zhang, 2015; Dong et al., 2014). Specifically, one or more aspects are present in a sentence, and aspects may express different sentiment polarities. The purpose of the task is to detect the sentiment polarities (i.e., POSITIVE, NEGATIVE, NEUTRAL) of all given aspects. Given the sentence "The **decor** is not a special at all but their amazing **food** makes up for it" and corresponding aspects "decor" and "food", the sentiment polarity towards "decor" is NEGATIVE, whereas the sentiment for "food" is POSITIVE.

Early works (Vo and Zhang, 2015; Kiritchenko et al., 2014; Schouten and Frasincar, 2016) to deal with ABSA mainly relied on manually designing syntactic features, which is cumbersome and ineffective as well. Subsequently, various neural network-based models (Kiritchenko et al., 2014; Vo and Zhang, 2015; Chen et al., 2017; Zhang et al., 2019b; Wang et al., 2020; Trusca et al., 2020) have been proposed to deal with ABSA tasks, to get rid of hand-crafted feature design. In these studies, syntactic structures proved effective, helping to connect aspects to the corresponding opinion words, thereby enhancing the effectiveness of the ABSA task (Zhang et al., 2019b; Tian et al., 2021; Veyseh et al., 2020; Huang and Carley, 2019; Sun et al., 2019; Wang et al., 2020). Additionally, some research (Chen et al., 2020a; Dai et al., 2021; Zhou et al., 2021; Chen et al., 2022; Brauwers and Frasincar, 2023) suggests there should exist task-specific induced latent structures because dependency syntactic structures (following that, we refer to them as external structures for convenience) generated by off-the-shelf dependency parsers are static and sub-optimal in ABSA. The syntactic structure is not specially designed to capture the interactions between aspects and opinion words.

Consequently, we classify these structure-based ABSA models into three categories by summarizing prior research: (1.) external structure, (2.) semi-induced structure, and (3.) full-induced structure. Works based on external structures use dependency syntactic structures generated by dependency parsers or modified dependency syntactic structures to provide structural support for ABSA (Zhang

---

*Corresponding author
[1]Our code is at https://github.com/hankniu01/FLT

et al., 2019b; Sun et al., 2019; Wang et al., 2020). Studies based on semi-induced structures leverage both external and induced structures, merging them to offer structural support for ABSA (Chen et al., 2020a). The first two categories require the introduction of external structures, which increases the complexity of preprocessing, while the third category directly eliminates this burdensomeness.

Our research is based on full-induced structures. Works in this field intend to totally eliminate the reliance on external structures to aid ABSA by employing pre-trained language models (PLMs) to induce task-specific latent structures (Dai et al., 2021; Zhou et al., 2021; Chen et al., 2022). These efforts, however, aim to create a tree-based structure, then convert it into a graph structure and feed it to Graph Neural Networks (GNNs) to capture structural information. Our research follows this line of thought, but directly from the perspective of the graph, utilizing PLMs to induce a graph structure for GNNs. In addition, studies (Tamkin et al., 2020) have shown that contextual representation contains information about context tokens as well as a wide range of linguistic phenomena, including constituent labels, relationships between entities, dependencies, coreference, etc. That is, there are various scales of information (spanning from the (sub)word itself to its containing phrase, clause, sentence, paragraph, etc.) in the contextual representation. This contextual representational characteristic has rarely been explored in previous studies. Therefore, our research investigates the influence of manipulations at informational scales of contextual representation on structure induction with spectral perspective.

Specifically, we employ graph structure learning (GSL) based on metric learning (Zhu et al., 2021) to induce latent structures from PLMs. We investigate three commonly used metric functions (Attention-based (Attn.), Kernel-based (Knl.), and Cosine-based (Cosine)) and contrast their effects on the structure of induced graphs. Furthermore, we heuristically explore four types of Frequency Filters with corresponding band allocations (HIGH, MID-HIGH, MID-LOW, LOW) acting on contextual representations, and in this way, we can segregate the representations of different scales at the level of individual neurons. Additionally, we introduce an automatic frequency selector (AFS) to circumvent the cumbersome heuristic approaches. This allows us to investigate the impact of manipulations at scale information for structure induction in contextual representations.

We employ three commonly PLMs: $BERT_{base}$, $RoBERTa_{base}$, $RoBERTa_{large}$. Our research is based on extensive experiments and yields some intriguing findings, which we summarize as follows:

**Structure Induction.** By comparing three GSL methods (Attention-based (Attn.), Kernel-based (Knl.), and Cosine-based (Cosine)), we find that the Attention-based method is the best for structure induction on ABSA.

**Frequency Filter (FLT).** Heuristic operations of information scales in the contextual representation by Frequency Filters are able to influence structure induction. Based on Attention-based GSL, the structure induction of FLT can obtain lower Aspects-sentiment Distance (AsD) and better performance.

**Automatic Frequency Selector (AFS).** Get rid of the tediousness of the heuristic method, AFS can consistently achieve better results than the Attention-based GSL method. This further demonstrates the effectiveness of manipulating scale information.

## 2 Related Work

### 2.1 Tree Induction for ABSA

In ABSA, there are a lot of works that aim to integrate dependency syntactic information into neural networks (Zhang et al., 2019b; Sun et al., 2019; Wang et al., 2020) to enhance the performance of ABSA. Despite the improvement of dependency tree integration, this is still not ideal since off-the-shelf dependency parsers are static, have parsing errors, and are suboptimal for a particular task. Hence, some effort is being directed toward dynamically learning task-specific tree structures for ABSA. For example, (Chen et al., 2020a) combines syntactic dependency trees and automatically induced latent graph structure by a gate mechanism. (Chen et al., 2022) propose to induce an aspect-specific latent tree structure by utilizing policy-based reinforcement learning. (Zhou et al., 2021) learn an aspect-specific tree structure from the perspective of closing the distance between aspect and opinion. (Dai et al., 2021) propose to induce tree structure from fine-tuned PLMs for ABSA. However, most of them fall to take the context representational characteristic into account.

## 2.2 Spectral Approach in NLP

In NLP, one line of spectral methods is used in the study of improving efficiency (Han et al., 2022; Zhang et al., 2018). For example, (Han et al., 2022) propose a new type of recurrent neural network with the help of the discrete Fourier transformer and gain faster training. In addition, a few works investigate contextual representation learning from the standpoint of spectral methods. (Kayal and Tsatsaronis, 2019) propose a method to construct sentence embeddings by exploiting a spectral decomposition method rooted in fluid dynamics. (Müller-Eberstein et al., 2022; Tamkin et al., 2020) propose using Frequency Filters to constrain different neurons to model structures at different scales. These bring new inspiration to the research of language representation.

## 2.3 Metric Learning based GSL

The metric learning approach is one of representative graph structure learning (GSL), where edge weights are derived from learning a metric function between pairwise representations (Zhu et al., 2021). According to metric functions, the metric learning approach can be categorized into two subgroups: Kernel-based and Attention-based. Kernel-based approaches utilize traditional kernel functions as the metric function to model edge weights (Li et al., 2018; Yu et al., 2020; Zhao et al., 2021b). Attention-based approaches usually utilize attention networks or more complicated neural networks to capture the interaction between pairwise representations (Velickovic et al., 2018; Jiang et al., 2019; Chen et al., 2020b; Zhao et al., 2021a). The Cosine-based method (Chen et al., 2020b) is generally a kind of Attention-based method. In our experiments, we take it out as a representative method.

## 3 Method

To obtain induced graph structure, we propose a spectral **filt**er (FLT) approach to select scale information when adaptively learning graph structure. In this section, we introduce a simple but effective approach (FLT) to induce graph structures from PLMs to enhance the performance of ABSA. The overall architecture is displayed in Figure 1.

### 3.1 Overview

As shown in Figure 1, the overall architecture is composed of PLMs, Graph Learner, GNNs architecture, and Prediction Head under normal cir-

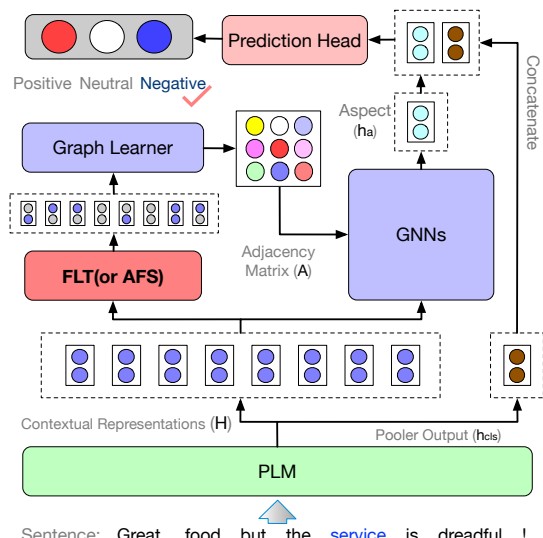

Figure 1: The overall architecture of our method.

cumstances. For a given input sentence $S = \{w_1, w_2, \cdots, w_n\}$, we employ a type of PLMs to serve as the contextual encoder to obtain the hidden contextual representation $\mathbf{H} \in \mathbb{R}^{n \times d}$ of the input sentence $S$, where $d$ is the dimension of word representations, and $n$ is the length of the given sentence. The contextual representation $\mathbf{H}$ is waited for inputting into GNNs architecture as node representations. Simultaneously, it is going to feed into Graph Learner to induce latent graph structures, which serve as adjacency matrices $\mathbf{A}$ for GNNs architecture. Then the GNNs architecture can extract aspect-specific features $\mathbf{h}_a$ utilizing both structural information from $\mathbf{A}$ and pre-trained knowledge information from $\mathbf{H}$. Finally, we concatenate the representation of [CLS] token $\mathbf{h}_{cls}$ from PLMs as well as $\mathbf{h}_a$, and send them into a Multi-layer Perception (MLP) (served as the Prediction Head) to detect the sentiment polarities (i.e., POSITIVE, NEGATIVE, NEUTRAL) for the given aspects.

Here, we investigate the effectiveness of three common graph structure learning (GSL) methods based on metric learning: Attention-based (Attn.), Kernel-based (Knl.), and Cosine-based (Cosine) (refer to (Zhu et al., 2021) for specific descriptions of Kernel-based and Cosine-based methods). We introduce the Attention-based GSL method to adaptively induce graph structures. Firstly, we calculate the unnormalized pair-wise edge score $e_{ij}$ for the $i$-th and $j$-th words utilizing the given representations $\mathbf{h}_i \in \mathbb{R}^d$ and $\mathbf{h}_j \in \mathbb{R}^d$. Specifically, the pair-wise edge score $e_{ij}$ is calculated as follows:

$$e_{ij} = (\mathbf{W}_i \mathbf{h}_i)(\mathbf{W}_j \mathbf{h}_j)^\top, \quad (1)$$

where $\mathbf{W}_i, \mathbf{W}_j \in \mathbb{R}^{d \times d_h}$ are learnable weights for $i$-th and $j$-th word representations, where $d_h$ is the hidden dimension.

Then, relying on these pair-wise scores $e_{ij}$ for all word pairs, we construct the adjacency matrices $\mathbf{A}$ for induced graph structures. Concretely,

$$\mathbf{A}_{ij} = \begin{cases} 1 & \text{if} \quad i = j \\ \frac{exp(e_{ij})}{\sum_{k=1}^{n} exp(e_{ik})} & \text{otherwise} \end{cases}, \quad (2)$$

where the adaptive adjacency matrix is $\mathbf{A} \in \mathbb{R}^{n \times n}$, and $\mathbf{A}_{ij}$ is the weight score of the edge between the $i$-th and $j$-th words.

For simplicity, we employ Vallina Graph Neural Networks (GCNs) (Kipf and Welling, 2017) served as GNNs architecture (other variants of graph neural networks can also be employed here). Given the word representations $\mathbf{H}$ and the adaptive adjacency matrix $\mathbf{A}$, we can construct an induced graph structure consisting of words (each word acts as a node in the graph) and feed it into GCNs. Specifically,

$$\mathbf{h}_i^l = \sigma(\sum_{j=1}^{n} \mathbf{A}_{ij} \mathbf{W}^l \mathbf{h}_j^{l-1} + \mathbf{b}^l), \quad (3)$$

where $\sigma$ is an activation function (e.g. ReLU), $\mathbf{W}^l$ and $\mathbf{b}^l$ are the learnable weight and bias term of the $l$-th GCN layer. By stacking several layers of Graph Learner and GNNs architectures, we can obtain structure information enhanced word representations $\mathbf{H}_g$ for the downstream task. It should be noted that the induced graph structure is dynamically updated while training.

After we get aspect representations $\mathbf{h}_a$ from $\mathbf{H}_g$, we feed them along with the pooler output $\mathbf{h}_{cls}$ of PLMs (the output representation of [CLS] token) into a task-specific Prediction Head to acquire results for the downstream task.

### 3.2 Frequency Filter (FLT)

Furthermore, inspired by (Tamkin et al., 2020), we introduce a spectral analysis approach to enhance the structure induction ability of the Graph Learner. Intuitively, we tend to import a Frequency Filter on contextual word representations to manipulate on scale information, and then feed them into the Graph Learner module to improve the structure induction capability. Contextual representations have been investigated to not only convey the meaning of words in context (Peters et al., 2018), but also carry a large range of linguistic information such

Table 1: Statistics of datasets.

| Dataset | Positive | | Neutral | | Negative | |
|---|---|---|---|---|---|---|
| | Train | Test | Train | Test | Train | Test |
| Rest14 | 2164 | 728 | 807 | 196 | 637 | 196 |
| Laptop14 | 994 | 341 | 870 | 128 | 464 | 169 |
| Twitter | 1561 | 173 | 3127 | 346 | 1560 | 173 |

as semantic roles, coreference, and constituent labels, etc. (Tenney et al., 2019). Prism (Tamkin et al., 2020) demonstrates these word representations contain multi-scale information ranging from (sub)word to phrase, clause, sentence, and so forth. Hence in this work, we explore the impact of structure induction ability by operating on scale-specific information of contextual representations.

To achieve this goal, we introduce a Frequency Filter (FLT) based on Discrete Fourier Transform (DFT) to conduct disentangling operations in the frequency domain. To be specific, given word representations $\mathbf{H} \in \mathbb{R}^{n \times d}$, we feed them into the FLT before the Graph Learner. For the specific $i$-th and $j$-th word representations $\mathbf{h}_i \in \mathbb{R}^d$ and $\mathbf{h}_j \in \mathbb{R}^d$, the pair-wise edge score $e_{ij}$ is calculated as follows:

$$\Phi^{flt}(x) = \mathcal{F}^{-1}\Big(\Psi\big(\mathcal{F}(x)\big)\Big), \quad (4)$$

$$e_{ij} = \Phi^{flt}(\mathbf{W}_i \mathbf{h}_i) \Phi^{flt}(\mathbf{W}_j \mathbf{h}_j)^{\top}, \quad (5)$$

where $\mathcal{F}(\cdot)$ and $\mathcal{F}^{-1}(\cdot)$ denote the Fast Fourier Transform (FFT) and its inverse, $\Psi$ indicates the filtering operation, and $\Phi^{flt}$ denotes the Frequency Filter (FLT). We carry out filtering at the sentence level. Subsequent operations are consistent with Section 3.1. We conduct experiments and analyses on four band allocations (HIGH, MID-HIGH, MID-LOW, LOW)). The specific band allocations are displayed in Table 5, and the analysis experiments refer to Section 4.7 and 4.10.

## 4 Experiment

To prove the effectiveness of our approach, we demonstrate experimental results conducted on three datasets for ABSA and compare them with previous works. We show the details as follows.

### 4.1 Dataset

We conduct experiments on SemEval 2014 task (Rest14 and Laptop14) (Pontiki et al., 2014) and Twitter (Dong et al., 2014) datasets, which are widely used. Each of the three datasets contains

Table 2: Overall performance of ABSA on the three datasets. According to the categorization of structure (Dep.: external structures (dependency syntactic tree), Semi.: semi-induced structures, Full: full-induced structures, and None: no structure information used), we classify the baselines accordingly, which are in the 'Structure' column.

| Embedding | Model | Structure | Rest14 | | Laptop14 | | Twitter | |
|---|---|---|---|---|---|---|---|---|
| | | | *Accuracy* | *Macro-F1* | *Accuracy* | *Macro-F1* | *Accuracy* | *Macro-F1* |
| Static Embedding | depGCN | Dep. | 80.77♯ | 72.02♯ | 75.55♯ | 71.05♯ | | |
| | CDT | Dep. | 82.30♯ | 74.02♯ | 77.19♯ | 72.99♯ | | |
| | kumaGCN | Semi. | 81.43 | 73.64 | 76.12 | 72.42 | 72.45 | 70.77 |
| | RGAT | Dep. | 83.30 | 76.08 | 77.42 | 73.76 | 75.57 | 73.82 |
| | FT-RoBERTa(ASGCN) | Full | 82.31 | 73.53 | 76.33 | 72.76 | 73.84 | 72.66 |
| | FT-RoBERTa(PWCN) | Full | 82.40 | 73.95 | 76.95 | 73.21 | 73.84 | 71.43 |
| | FT-RoBERTa(RGAT) | Full | 82.76 | 75.25 | 77.43 | 74.21 | 75.43 | 74.04 |
| BERT$_{base}$ | BERT | None | 85.62♯ | 78.28♯ | 77.58♯ | 72.38♯ | 75.28 | 74.11 |
| | SAGAT | Dep. | 85.08 | 77.94 | 80.37 | 76.94 | 75.40 | 74.17 |
| | DGEDT | Dep. | 86.30 | 80.00 | 79.80 | 75.60 | 77.90 | 75.40 |
| | depGCN-BERT | Dep. | 85.00 | 78.79 | 81.19 | 77.67 | 75.58 | 74.58 |
| | RGAT-BERT | Dep. | 86.60 | 81.35 | 78.21 | 74.07 | 76.15 | 74.88 |
| | KumaGCN-BERT | Semi. | 86.43 | 80.30 | 81.98 | 78.81 | 77.89 | 77.03 |
| | dotGCN-BERT | Full | 86.16 | 80.49 | 81.03 | 78.10 | **78.11** | 77.00 |
| RoBERTa$_{base}$ | Roberta + MLP | None | 87.32 | 81.01 | 82.60 | 79.33 | 77.17 | 76.20 |
| | RoBERTa-ASC(Dep) | Dep. | 82.82 | 75.12 | 74.12 | 70.52 | - | - |
| | LCFS-ASC-CDW(Dep) | Dep. | 86.71 | 80.31 | 80.52 | 77.13 | - | - |
| | Dep(ASGCN) | Dep. | 86.90 | 80.75 | 81.66 | 78.31 | 75.28 | 74.38 |
| | Dep(PWCN) | Dep. | 87.41 | 81.07 | 84.16 | 81.18 | 76.63 | 75.60 |
| | Dep(RGAT) | Dep. | 87.43 | 80.61 | 83.43 | 80.28 | 74.42 | 72.93 |
| | FT-RoBERTa(ASGCN) | Full | 86.87 | 80.59 | 83.33 | 80.32 | 76.10 | 75.07 |
| | FT-RoBERTa(PWCN) | Full | 87.35 | 80.85 | 84.01 | 81.08 | 77.02 | 75.52 |
| | FT-RoBERTa(RGAT) | Full | 87.52 | 81.29 | 83.33 | 79.95 | 75.81 | 74.91 |
| | **FLT** | Full | 88.57 | 83.27 | 85.42 | 83.01 | 77.02 | 75.83 |
| RoBERTa$_{large}$ | **FLT** | Full | **90.27** | **85.20** | **86.05** | **84.68** | 77.89 | **77.20** |

three sentiment label categories: POSITIVE, NEUTRAL, and NEGATIVE. Statistics of these datasets are displayed in Table 1, where (Train|Test) denotes the number of instances on the training, and testing set for each dataset.

## 4.2 Experiment Settings

We utilize the popular Pre-trained Language Models (PLMs) based on Transformer Encoder architecture (BERT$_{base}$ (Devlin et al., 2019), RoBERTa$_{base}$ and RoBERTa$_{large}$ (Liu et al., 2019)) for word representations. Moreover, the hidden dimensions of all Graph Learners are 60. The dropout rate is 0.2, the batch size is 32. The number of the epoch is 60 for RoBERTa$_{base}$ and RoBERTa$_{large}$, and 30 for BERT$_{base}$. We use Adam optimizer (Kingma and Ba, 2015) while training with the learning rate initialized by 1e-5. Following previous works, we use Accuracy and Macro-F1 scores for metrics. All experiments are conducted on NVIDIA Tesla P100.

## 4.3 Baselines

We categorize the existing structure-based ASBA models into three genres: external structure, semi-induced structure, and full-induced structure. Below, we introduce each of them in detail.

**External Structure.** This line of works utilizes dependency syntactic structure generated by external dependency parsers (e.g. Spacy and Standford CoreNLP [2], etc.) to offer structural information supplements for ABSA. Its delegate works as follows:

**depGCN** (Zhang et al., 2019a) combines BiL-STM to capture contextual information regarding word orders with multi-layered GCNs.

**CDT** (Sun et al., 2019) encodes both dependency and contextual information by utilizing GCNs and BiLSTM.

**RGAT** (Wang et al., 2020) feeds reshaped syntactic dependency graph into RGAT to capture aspect-centric information.

**SAGAT** (Huang et al., 2020) uses graph attention network and BERT to explore both syntax and semantic information for ABSA.

**DGEDT** (Tang et al., 2020) jointly consider BERT outputs and dependency syntactic representations by utilizing GCNs.

**LCFS-ASC-CDW** (Phan and Ogunbona, 2020) combine dependency syntactic embeddings, part-of-speech embeddings, and contextualized embeddings to enhance the performance of ABSA.

---

[2]https://stanfordnlp.github.io/CoreNLP/

Table 3: Results of ablation studies.

| Embedding | Model | Structure | Rest14 | | Laptop14 | | Twitter | |
|---|---|---|---|---|---|---|---|---|
| | | | Accuracy | Macro-F1 | Accuracy | Macro-F1 | Accuracy | Macro-F1 |
| BERT$_{base}$ | Attn. | Full | 85.43 | 78.04 | 80.54 | 77.06 | 76.22 | 75.04 |
| | **FLT** | Full | **87.04** | **81.46** | **81.17** | **77.97** | **77.55** | **76.66** |
| RoBERTa$_{base}$ | Attn. | Full | 87.59 | 81.72 | 83.86 | 80.53 | 75.72 | 73.92 |
| | **FLT** | Full | **88.57** | **83.27** | **85.42** | **83.01** | **77.02** | **75.83** |
| RoBERTa$_{large}$ | Attn. | Full | 89.46 | 84.12 | 84.80 | 82.19 | 77.02 | 75.75 |
| | **FLT** | Full | **90.27** | **85.20** | **86.05** | **84.68** | **77.89** | **77.20** |

**Semi-induced Structure.** Works in this line tend to exploit both dependency syntactic structure from off-the-shelf parsers and induced structure from PLMs, the representative works are as follows:

**KumaGCN** (Chen et al., 2020a) combine latent graphs induced by self-attention neural networks and dependency syntactic structure for ABSA.

**Full-induced Structure.** Works in this line intend to get totally rid of external parsers and induce task-specific latent structures from PLMs for downstream tasks. Its delegate works as follows:

**FT-RoBERTa** (Dai et al., 2021) induce tree structures from the fine-tuned RoBERTa (fine-tune RoBERTa on the ABSA datasets in advance) by utilizing a dependency probing approach.

**dotGCN** (Chen et al., 2022) induce aspect-specific opinion tree structures by using Reinforcement learning and attention-based regularization.

### 4.4 Overall Performance

The overall results of competitive approaches and FLT on the three benchmarks are shown in Table 2. We categorize baselines according to the embedding type (static embedding (GloVe), BERT$_{base}$, RoBERTa$_{base}$, and RoBERTa$_{large}$) and the structure they used (None, Dep., Semi., and Full). The parameters of PLMs are trained together with the GSL module for FLT. Compared with baselines, FLT obtains the best results except on Twitter, which obtains comparable results. We speculate that the reason is that the expression of Twitter is more casual, which leads to a limited improvement of the structure on Twitter, which is consistent with the result in (Dai et al., 2021). Compared with FT-RoBERTa-series works, the most relevant work of ours, FLT outperforms them a lot on the three datasets. And it is worth noting that FT-RoBERTa-series works need fine-tuning PLMs on the ABSA datasets in advance (Dai et al., 2021), but FLT does not need it. Therefore, FLT is simpler and more effective than FT-RoBERTa-series works.

Table 4: The impact of different metric functions based on RoBERTa$_{base}$.

| Metric | Rest14 | | Laptop14 | | Twitter | |
|---|---|---|---|---|---|---|
| | Accuracy | Macro-F1 | Accuracy | Macro-F1 | Accuracy | Macro-F1 |
| Attn. | 87.59 | 81.72 | 83.86 | 80.53 | 75.72 | 73.92 |
| Knl. | 87.14 | 80.45 | 83.54 | 80.44 | **76.01** | **73.98** |
| Cosine | 87.14 | 79.94 | 83.39 | 79.93 | 74.28 | 72.80 |

### 4.5 Ablation Study

We conduct ablation studies to highlight the effectiveness of FLT, which is based on Attention-based (Attn.) GSL module and utilizing Frequency Filter. Thus, we compare Attn. and FLT on three PLMs (BERT$_{base}$, RoBERTa$_{base}$, and RoBERTa$_{large}$) to show the impact of introducing Frequency Filter. Results are shown in Table 3. Compared to Attn., FLT has achieved significant improvements in consistency across three datasets utilizing different PLMs. Therefore, it can be seen that the manipulation of scale information is beneficial for enhancing performance.

### 4.6 Different Metric Function

In this section, we contrast the impact of three representative metric functions: Attention-based (Attn.), Kernel-based (Knl.), and Cosine-based (Cosine) on structure induction. From the insight of graph structure learning (Chen et al., 2020b; Zhu et al., 2021), the common options for metric learning include attention mechanism (Vaswani et al., 2017; Jiang et al., 2019), radial basis function kernel (Li et al., 2018; Yeung and Chang, 2007), and cosine similarity (Wojke and Bewley, 2018). We follow these previous works to implement the counterpart metric functions (Knl. and Cosine) for comparison, the results are shown in Table 4. The performance of attention-based (Attn.) on the three benchmarks gains the best results except on Twitter. But the margin between Attn. and Knl. is not big (0.29% for Accuracy and 0.06% for Macro-F1) on Twitter, thus we select the metric function Attn. for later analysis.

Table 5: The spectral bands we consider in this work. Since the task considered in this work is at the sentence level, we only take the scale from word to sentence into account. Here, $L$ denotes the sentence's length.

| Band | Scale | Period(Toks) | DFT index |
|---|---|---|---|
| HIGH | Word | $1 \to 2$ | $L/2 \to L$ |
| MID-HIGH | Phrase | $2 \to 6$ | $L/6 \to L/2$ |
| MID-LOW | Clause | $6 \to 14$ | $L/14 \to L/6$ |
| LOW | Sentence | $14 \to L$ | $1 \to L/14$ |

## 4.7 Different Frequency Filters

Table 6: Band impact based on RoBERTa$_{base}$. There are statistical results for heuristic frequency selection, and the results follow the form mean(standard deviation).

| Filter | Rest14 | | Laptop14 | | Twitter | |
|---|---|---|---|---|---|---|
| | Accuracy | Macro-F1 | Accuracy | Macro-F1 | Accuracy | Macro-F1 |
| HIGH | 87.54(0.55) | **81.33(0.97)** | 84.21(0.43) | 81.50(0.57) | 75.83(0.34) | 74.76(0.42) |
| MID-HIGH | **87.55(0.53)** | 81.31(1.06) | **84.39(0.78)** | **81.69(0.95)** | 75.71(0.78) | 74.68(0.72) |
| MID-LOW | 87.23(0.27) | 81.15(0.71) | 83.74(0.52) | 81.00(0.85) | **76.73(0.23)** | **75.64(0.12)** |
| LOW | 87.37(0.32) | 80.75(0.45) | 83.49(0.15) | 80.60(0.15) | 76.16(0.20) | 74.94(0.19) |

Figure 2: The distribution of sentence length on datasets (we combine training and testing sets for this statistic).

This section analyzes the impact of four different spectral bands (HIGH, MID-HIGH, MID-LOW, LOW) on structure induction. Each band reflects a diverse range of linguistic scales from word level to sentence level, the detailed setting is shown in Table 5. The different spectral bands are revealed by their period: the number of tokens it takes to complete a cycle. For example, the word scale suggests the period of $1 \to 2$ tokens, thus the spectral band should be $L/2 \to L$ if the sentence's length denotes $L$.

Then, we conduct analysis experiments on the three datasets to explore the impact of different spectral bands. The length $L$ in our experiments is 100, which fits the length distribution of all samples in these datasets. We perform multiple frequency selections in different frequency bands heuristically, and the performance of our model in different frequency bands on the three datasets is summarized in Table 6. Please refer to Appendix A for

the detailed frequency selection and results. Our model performs better in HIGH and MID-HIGH bands on Rest14 and Laptop14 but performs better in LOW and MID-LOW bands on Twitter. Combined with Figure 2, we find that the distribution of sentence length in Twitter is very distinct from that of Rest14 and Laptop14, the sentences in Twitter are generally longer, which leads to the fact that the clause- and sentence-scale information is more beneficial to the effect improvement.

## 4.8 Aspects-sentiment Distance

To illustrate the effectiveness of induced structure, following (Dai et al., 2021), we introduce the Aspects-sentiment Distance (AsD) to quantify the average distance between aspects and sentiment words in the induced structure. The AsD is calculated as follows:

$$C^\star = S_i \cap C, \tag{6}$$

$$AsD(S_i) = \frac{\sum\limits_A^{a_p} \sum\limits_{C^\star}^{c_q} dist(a_p, c_q)}{|A||C^\star|}, \tag{7}$$

$$AsD(D) = \frac{\sum\limits_D AsD(S_i)}{|D|}, \tag{8}$$

where $C = \langle c_1, \cdots c_q \rangle$ is a sentiment words set (following the setting from (Dai et al., 2021)), $S_i$ denotes each sentence in dataset $D$, and $A = \langle a_1, \cdots, a_p \rangle$ denotes the set of aspects for each sentence. We utilize $dist(n_1, n_2)$ to calculate the relative distance between two nodes ($n_1$ and $n_2$) on the graph structure, and $|\cdot|$ represent the number of elements in the given set. For a detailed setting, please refer to Appendix B.

The results are displayed in Table 7, and the less magnitude indicates the shorter distance between aspects and sentiment words. Compared to dependency structure (Dep.), attention-based GSL (Attn), and our method (FLT) shorten the Aspects-sentiment Distance greatly, which shows that GSL method encourages the aspects to find sentiment words. Furthermore, in comparison with Attn., FLT has a lower AsD score, which proves a reasonable adjustment on the scale level can obtain better structures.

## 4.9 Structure Visualization and Case Study

**Structure Visualization.** As shown in Figure 3, we visualize the difference of distinct structures: (a) is from the Spacy parser, (b) is from Attn., and (c)

| Structure | Rest14 | Laptop14 | Twitter |
|-----------|--------|----------|---------|
| Dep.      | 8.19   | 8.02     | 8.33    |
| Attn.     | 2.26   | 2.55     | 2.64    |
| FLT       | **1.97** | **2.15** | **2.16** |

Table 7: The Aspects-sentiment Distance (AsD) of different trees in all datasets. The dependency tree structure (Dep.) comes from the Spacy parser [3].

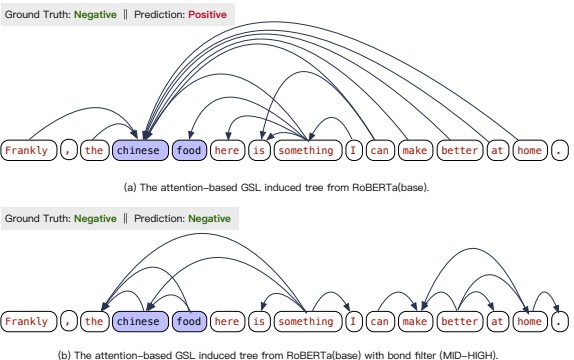

(a) The attention–based GSL induced tree from RoBERTa(base).

(b) The attention–based GSL induced tree from RoBERTa(base) with bond filter (MID–HIGH).

Figure 4: A case of Rest14 dataset. The colored words denote aspects. The golden label for *Chinese food* is NEGATIVE.

Table 8: The results of AFS based on RoBERTa$_{base}$.

| Model | Rest14 | | Laptop14 | | Twitter | |
|-------|--------|--------|----------|--------|---------|--------|
|       | *Accuracy* | *Macro-F1* | *Accuracy* | *Macro-F1* | *Accuracy* | *Macro-F1* |
| Attn. | 87.59 | 81.72 | 83.86 | 80.53 | 75.72 | 73.92 |
| **AFS** | **88.30** | **82.89** | **84.48** | **81.63** | **76.16** | **75.20** |

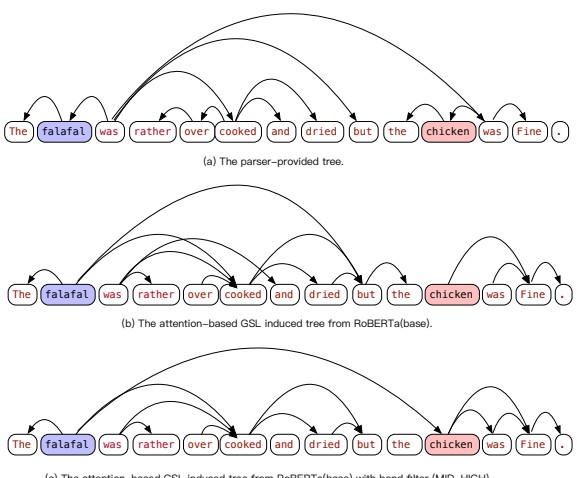

(a) The parser–provided tree.

(b) The attention–based GSL induced tree from RoBERTa(base).

(c) The attention–based GSL induced tree from RoBERTa(base) with bond filter (MID–HIGH).

Figure 3: A case is from the Rest14 dataset. The colored words are aspects. The golden label for *falafal* is NEGATIVE, and for *chicken* is POSITIVE.

is the result from FLT. This case is from the Rest14 dataset. In comparison with (a), aspects are more directly connected to important sentiment words (e.g. cooked, dried, and fine) in (b) and (c), which is consistent with the results of AsD in Section 4.8. In this case, both (b) and (c) obtained correct judgment results, hence from the perspective of structure, they are relatively similar.

**Case Study.** In Figure 4, we provide a case to compare Attn. in (a) and FLT in (b). In this case, the structures induced by the two are quite different, and for the aspect (*Chinese food*), Attn. gives a wrong judgment. From the comparison of structures, it can be found that although the aspect word *Chinese* in (a) pays attention to the key information *I can make better at home*, they may not understand the semantics expressed by this clause. From the perspective of structure, FLT in (b) is obviously better able to understand the meaning of this clause.

### 4.10 Automatic Frequency Selector (AFS).

Furthermore, in order to illustrate the impact of the operation of the scale information on the GSL, we introduce an Automatic Frequency Selector (AFS) to select helpful frequency components along with

the optimization of the overall model. In this way, for different datasets, the information of the corresponding scale (HIGH, MID-HIGH, etc.) can be adaptively selected to improve the effect of structure induction. Here we briefly describe the AFS, and for a detailed description, please refer to Appendix C.

**Model Description.** Following the operation of FLT, for an input sentence representation $\mathbf{H} \in \mathbb{R}^{n \times d}$, we conduct Discrete Fourier Transform (DFT) $\mathcal{F}$ to transform $\mathbf{H}$ into the frequency domain. Then, we utilize AFS $\Phi^{auto}$ to adaptively select frequency components, where AFS $\Phi^{auto}$ is realized by using a Multi-layer Perceptron (MLP) architecture, please refer to the Appendix C for details. After AFS and inverse Discrete Fourier Transform $\mathcal{F}^{-1}$, we can obtain the sentence representation $\mathbf{H}^{afs} \in \mathbb{R}^{n \times d}$. The subsequent operations are consistent with the attention-based GSL.

**Results.** We utilize AFS instead of FLT to conduct experiments on the three datasets, the results are shown in Table 8. Compared to Attn., AFS is consistently improved. This further illustrates the operation of scale information is conducive to improving the effectiveness of GSL on ABSA. Compared with the heuristic FLT method, AFS avoids the burden brought by manual frequency selection, making the method more flexible.

**Frequency Component Analysis.** Furthermore, we conducted an in-depth analysis of the intermedi-

Table 9: Frequency Component Analysis. The spectral bands we consider in this work. Since the task considered in this work is at the sentence level, we only take the scale from word to sentence into account. Here, $L$ denotes the sentence's length.

| Band | Rest14(%) | Laptop14(%) | Twitter(%) | Scale | DFT index |
|---|---|---|---|---|---|
| HIGH | 84.77 | 25.64 | 87.22 | Word | $L/2 \rightarrow L$ |
| MID-HIGH | 89.82 | 28.68 | 92.61 | Phrase | $L/6 \rightarrow L/2$ |
| MID-LOW | 91.82 | 41.02 | 96.87 | Clause | $L/14 \rightarrow L/6$ |
| LOW | 99.61 | 88.08 | 99.19 | Sentence | $1 \rightarrow L/14$ |
| Overall | 88.88 | 35.21 | 91.41 | - | - |

ate results obtained from the Automatic Frequency Selector (AFS). From Table 8, we observe that incorporating AFS consistently enhances model performance without manual adjustments to Frequency Components. This suggests that the automated Frequency Components selection process is effective. Based on AFS's Frequency Component selection outcomes, we performed statistical analyses across three datasets in accordance with the spectral band distribution outlined in Table 5. Table 9 illustrates the percentage of Frequency Components selected by AFS within different spectral bands, while "Overall" represents the percentage of selected Frequency Components across all four bands.

It is evident that the results are not uniformly 100%, indicating that AFS indeed performs selection on Frequency Components, thereby adjusting information at various scales to achieve consistent improvements. Moreover, the percentage of selected Frequency Components varies across different datasets, implying adaptive adjustments by AFS to cater to the diverse demands of distinct samples. Notably, the LOW band exhibits the highest percentage of selected Frequency Components, underscoring the significance of sentence-level information for token-level tasks (such as Structure Induction for ABSA, which can be considered a token-level task). This observation also aligns with the conclusion drawn in reference (Müller-Eberstein et al., 2022).

## 5   Conclusion

In this work, we propose utilizing GSL to induce latent structures from PLMs for ABSA and introduce spectral methods (FLT and AFS) into this problem. We also explore the impact of manipulation on scale information of the contextual representation for structure induction. Extensive experiments and analyses have demonstrated that the operation of scale information of contextual representation can enhance the effect of GSL on ABSA. Additionally, our exploration is also beneficial to provide inspiration for other similar domains.

## Limitations

Though we verify the operation on various information scales can be beneficial to structure induction on ABSA, there are still some limitations. Although the heuristic FLT has achieved excellent results, it requires some manual intervention. The AFS method reduces manual participation, but its effect is worse than the optimal FLT method. However, it is still meaningful to explore the impact of scale information on the contextual representation of downstream tasks.

## Acknowledgements

This work is funded in part by the National Natural Science Foundation of China Project (No.U1936213), and the Major Key Project of PCL (PCL2021A06).

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

## A Different Frequency Selection

We heuristically select spectral bands (HIGH, MID-HIGH, MID-LOW, LOW) to observe the impact of different spectral bands on structure induction for ABSA. The specific selection of spectral bands at different frequencies and their results are shown in Table 10. The range of spectral bands corresponds to the description in Table 5. Here, based on the distribution of sentence lengths in the dataset (refer to Figure 2), we set the maximum length (L) to

100 for each dataset and place sentences of similar length in one batch, with a batch size of 32. Each batch is batched according to the maximum sentence length in that batch. For simplicity, we did not design specific spectral bands for different sentence lengths. Instead, we set the spectral bands based on the maximum sentence length (L) in each dataset. We only change the hyperparameter 'Bands' settings, while all other settings remain the same. For specific experimental settings, refer to Section 4.2. It can be observed that different spectral band selections indeed lead to different results, and an appropriate heuristic spectral band selection can significantly improve the results.

## B The Settings of AsD analysis

Here, we provide a detailed introduction to the relative distance calculation $dist(n_1, n_2)$ for AsD. For a given sentence $S_i$, with its aspect words $A = \langle a_1, \cdots, a_p \rangle$, sentiment word set $C = \langle c_1, \cdots c_q \rangle$, and the adjacency matrix $A_G$ of the induced graph structure, we calculate the shortest hops from $a_p$ to $c_q$. If the value of the corresponding position of $a_p$ and $c_q$ on the adjacency matrix $A_G$ is greater than the threshold $\gamma$, then we call the distance between $a_p$ and $c_q$ to be 1. Otherwise, finding the shortest hops between $a_p$ and $c_q$ on the $A_G$ as its shortest path. We also use $\gamma$ to judge whether there is an edge between two nodes. Here, $\gamma$ is set to the average value of all values of $A_G$. If $a_p$ and $c_q$ are not directly connected, we set the distance between $a_p$ and $c_q$ to the maximum number of hops, where the maximum number of hops is set to 10.

## C Automatic Frequency Selector (AFS)

Furthermore, it is not affirmed that information in just one band (e.g. HIGH, MID-HIGH, etc.) is helpful, and information in other bands may also provide a gaining effect. Therefore with this in mind, we introduce an Automatic Frequency Selector (AFS) to select helpful frequency components along with the optimization of the overall model.

To achieve this goal, we design the Frequency Selection operation under a probabilistic scenario $\Upsilon$. To be specific, we map each frequency component $f$ into a Bernoulli parameter space by employing a Multi-layer Perceptron (MLP) architecture to parameterize this mapping process. Firstly, we bring in a set of learnable parameters $\xi \in \mathbb{R}^{k \times d_k}$ to parameterize frequency components, where $k$ denotes the number of frequency components, and $d_k$ de-

notes the dimension of component representations. Then, we utilize the MLP architecture (composed of two linear projection layers $Proj_1$ and $Proj_2$, and an activation function $\sigma$ (i.e. ReLU)) to map frequency components $\xi$ into the Bernoulli parameter space.

$$z_B = MLP(\xi) = Proj_2\Big(\sigma\big(Proj_1(\xi)\big)\Big), \quad (9)$$

$$\xi_B = \varphi\Big(\big(z_B - log\big(-log(\epsilon)\big)\big)/\tau\Big) \quad (10)$$

where $\xi_B$ denotes the success probabilities of Bernoulli distributions, and $\varphi$ denotes the Softmax function. We utilize the Gumbel reparameterization proposed by (Jang et al., 2017; Maddison et al., 2017) to address the differentiable difficulty during training, where $\epsilon \sim \mathcal{U}(0,1)$ is random variables of a uniform distribution on the interval $(0,1)$. The hyperparameter $\tau \to 0$ is the annealing temperature, which is adjusted to zero progressively in practice. Next, we can obtain the values of Bernoulli random variables $m_B \sim Bern(\xi_B)$, where $m_B \in \{0,1\}^k$, and $Bern$ denotes Bernoulli distributions. During the non-training phase, we set a hyperparameter threshold $\gamma$ to control the sparsity of $m_B$. (For the Rest14 dataset, we set the threshold $\gamma$ to 0.65. For the other two datasets, the threshold is set at 0.75.)

Subsequently, for the $i$-th and $j$-th word representations $\mathbf{h}_i \in \mathbb{R}^d$ and $\mathbf{h}_j \in \mathbb{R}^d$, we can calculate the pair-wise edge score $e_{ij}$ as follows:

$$\Phi^{afs}(x) = \mathcal{F}^{-1}\Big(\Upsilon\big(\mathcal{F}(x)\big)\Big), \quad (11)$$

$$e_{ij} = \Phi^{afs}(\mathbf{W}_i\mathbf{h}_i)\Phi^{afs}(\mathbf{W}_j\mathbf{h}_j)^{\top}, \quad (12)$$

where $\Upsilon$ indicates the Frequency Selection operation, and $\Phi^{afs}$ denotes the Automatic Frequency Selector (AFS). Subsequent operations are consistent with Section 3.1.

Table 10: Detailed results of the band impact based on RoBERTa$_{base}$ for heuristic frequency selection. For real sequence, the spectrum obtained by the Discrete Fourier Transform is symmetrical, so we only take half of the spectral bands for analysis. Negative values indicate that the frequency is selected from the high-frequency band, and positive values mean that the frequency is selected from the low-frequency band. Additionally, $x \rightarrow y$ means that the frequency selection is between the two values ($x$ and $y$). The values in **bold** indicate superior performance compared to the Attn. method.

| Filter | Bands | Rest14 | | Laptop14 | | Twitter | |
|---|---|---|---|---|---|---|---|
| | | *Accuracy* | *Macro-F1* | *Accuracy* | *Macro-F1* | *Accuracy* | *Macro-F1* |
| HIGH | -1 | 87.32 | 80.76 | **84.48** | **81.54** | 75.43 | **74.88** |
| | -2 | 87.32 | 80.79 | **84.17** | **81.13** | **75.72** | **74.45** |
| | -3 | 87.23 | 81.56 | **83.86** | **81.20** | **76.01** | **74.34** |
| | -4 | 86.88 | 80.44 | **84.01** | **81.34** | 75.43 | **74.78** |
| | -5 | **87.77** | 81.62 | 83.54 | **80.53** | **76.30** | **75.00** |
| | -6 | **87.77** | 81.71 | 82.76 | 79.93 | 75.58 | **74.34** |
| | -8 | **87.77** | 80.74 | **84.80** | **82.27** | **76.16** | **75.52** |
| | -10 | 87.05 | 80.79 | **83.86** | **81.37** | 75.58 | **74.41** |
| | -12 | **87.77** | 80.74 | **84.48** | **81.38** | **75.87** | **74.45** |
| | -14 | **87.75** | **81.86** | **84.80** | **82.21** | 75.43 | **74.69** |
| | -16 | **88.57** | **82.95** | **84.32** | **81.87** | **76.45** | **75.46** |
| | -18 | 86.43 | 79.26 | 83.54 | **80.54** | 75.43 | **74.08** |
| | -20 | **88.13** | **82.33** | **84.01** | **81.06** | **76.01** | **75.23** |
| | -22 | **88.57** | **83.27** | **84.48** | **81.82** | 75.58 | **74.91** |
| | -24 | 87.14 | 80.63 | **84.17** | **81.65** | **76.30** | **75.18** |
| | -26 | 87.50 | 80.85 | **84.64** | **82.04** | **76.01** | **74.46** |
| MID-HIGH | $8 \rightarrow 10$ | **88.21** | **82.41** | **84.48** | **81.90** | 74.57 | 74.19 |
| | $8 \rightarrow 11$ | **87.86** | 81.69 | **85.42** | **83.01** | 75.29 | **74.59** |
| | $8 \rightarrow 12$ | 87.50 | 80.66 | 83.39 | 80.49 | 75.29 | **74.68** |
| | $8 \rightarrow 13$ | 87.23 | 80.13 | **83.86** | **81.06** | **76.88** | **75.70** |
| | $8 \rightarrow 14$ | 86.88 | 80.75 | **84.48** | **81.70** | **75.72** | **74.90** |
| | $8 \rightarrow 16$ | **87.95** | 81.69 | 83.70 | **80.92** | **77.02** | **75.84** |
| | $8 \rightarrow 18$ | 87.50 | **82.16** | **85.27** | **82.67** | **75.72** | **74.48** |
| | $8 \rightarrow 20$ | **88.48** | **83.32** | 83.70 | **80.81** | 75.14 | 73.63 |
| | $8 \rightarrow 22$ | 87.05 | 79.81 | 83.54 | 80.50 | **76.45** | **75.16** |
| | $8 \rightarrow 24$ | 86.88 | 80.53 | **84.33** | **81.65** | 75.00 | 73.62 |
| MID-LOW | $4 \rightarrow 5$ | 86.96 | 80.50 | **84.01** | **81.14** | **76.45** | **75.50** |
| | $4 \rightarrow 6$ | 87.14 | 80.40 | 83.70 | **81.05** | **76.59** | **75.61** |
| | $4 \rightarrow 7$ | 87.14 | 81.71 | **84.33** | **82.10** | **77.02** | **75.64** |
| | $4 \rightarrow 8$ | **87.68** | **81.99** | 82.92 | 79.72 | **76.87** | **75.82** |
| LOW | 1 | 87.41 | 81.27 | 83.39 | 80.44 | **76.16** | **75.03** |
| | 2 | **87.86** | 81.06 | 83.39 | **80.55** | **76.15** | **75.16** |
| | 3 | 87.23 | 80.51 | 83.70 | **80.80** | **76.45** | **74.90** |
| | 4 | 86.96 | 80.14 | **84.01** | **81.64** | **75.87** | **74.65** |
| Attn. | - | 87.59 | 81.72 | 83.86 | 80.53 | 75.72 | 73.92 |