# OpenReview forum: "Adaptive Structure Induction for Aspect-based Sentiment Analysis with Spectral Perspective"
_EMNLP/2023/Conference — EMNLP 2023 Findings_

### Official Review · Reviewer_sCU9 · 2023-07-25

**Soundness:** 3

**Excitement:**

4: Strong: This paper deepens the understanding of some phenomenon or lowers the barriers to an existing research direction.

**Justification For Ethical Concerns:**

None.

**Missing References:**

How does approach compare to hybris solutions as HAABSA++:

Maria Mihaela Trusca, Daan Wassenberg, Flavius Frasincar, Rommert Dekker: A Hybrid Approach for Aspect-Based Sentiment Analysis Using Deep Contextual Word Embeddings and Hierarchical Attention. ICWE 2020: 365-380
?

As a paper on ABSA and ASBC, these two surveys need to be referred to:

-For ABSA:

Kim Schouten, Flavius Frasincar: Survey on Aspect-Level Sentiment Analysis. IEEE Trans. Knowl. Data Eng. 28(3): 813-830 (2016)

-For ABSC:

Gianni Brauwers, Flavius Frasincar: A Survey on Aspect-Based Sentiment Classification. ACM Comput. Surv. 55(4): 65:1-65:37 (2023)

**Paper Topic And Main Contributions:**

The paper presents a solution for ABSC based on fully-automatic Graph Structure Learning (GSL) using PLMs. The graphs are induced using three similarities: attention, kernel, and cosine. In addition a Frequency Filter (FLT) based on DFT selects frequencies from 5 bandwidths. Alternatively, an Automatic Filter Selector (AFS) helps with the automatic selection of the frequencies. Using three datasets and an ablation experiment the authors show the usefulness of the proposed approach and its components.

**Questions For The Authors:**

1. What is the relation between frequency and context? How is this related to DFT?
2. Did you try to combine multiple graphs based on the previous similarity functions and then using a fusion gate?
3. The comparison between FLT and AFS is not clear, as in the paper FLT is claimed to be the best while in conclusions AFS is claimed to be the best w.r.t. to the accuracy. What is correct?

Thank you for the answers to my questions. I am satisfied with the answers.

**Reasons To Accept:**

+ technical paper
+ good results
+ novel solution

**Reasons To Reject:**

- sometimes not clear explanations
- sometimes not clear results
- missing relevant literature

**Reproducibility:**

3: Could reproduce the results with some difficulty. The settings of parameters are underspecified or subjectively determined; the training/evaluation data are not widely available.

**Reviewer Confidence:**

5: Positive that my evaluation is correct. I read the paper very carefully and I am very familiar with related work.

**Typos Grammar Style And Presentation Improvements:**

-throughout the paper: all bold text followed by a paragraph needs a dot at the end (e.g., “External Structure.” Instead of “External Structure”)
-throughout the paper: “e.g.,” instead of “e.g.”
-page 1: “etc.” instead of “etc”
-page 3: “a simple” instead of “the simple”
-page 4: h_{CLS instead of h_cls
-page 5: “CoreNLP” instead of “Corenlp”
-page 5: “uses graph” instead of “use graph”
-page 6: “to users” instead of “of users”
-page 7: “mean (” instead of “mean(”
-page 7: “our method” instead of “Our method”
-page 9: “ABSA” instead of “ABAS”

---

> ### Author Rebuttal · Authors · 2023-08-29
>
> Dear reviewer, thank you so much for your detailed comments and affirmation of our approach.
>
> ## 1: Relation between Frequency, DFT and Context
>
> ### Related Works and Motivation
>
> Drawing from previous research ([1],[2]), it is well-established that contextual representations with varying scales of information in pre-trained models exert distinct influences on downstream tasks. For instance, reference [1] demonstrates the advantage of utilizing low-frequency signals for topic classification (document-level), while medium-frequency signals prove more beneficial for tasks such as dialog acts classification (utterance-level). Hence, it can be inferred that operations on different scale information within context representations may impact task performance downstream.
>
> In this study, we concentrate on conducting Structure Induction from contextual representations of pre-trained models. We apply the induced structure to tasks like Aspect-based Sentiment Analysis (ABSA) that require structural information. Our focus is on exploring whether operations concerning scale information are conducive to enhancing the efficacy of induced structures, thus yielding benefits for downstream tasks. To achieve this objective, segregating and extracting information of varying frequencies becomes imperative. This naturally leads us to consider Discrete Fourier Transform (DFT) as a spectral analysis tool, aligning with the methodology employed in prior research endeavors.
>
> [1] Alex Tamkin, Dan Jurafsky, and Noah D. Goodman. 2020. Language through a prism: A spectral approach for multiscale language representations. NeurIPS 2020, December 6-12, 2020, virtual.
>
> [2] Max Müller-Eberstein, Rob van der Goot, and Barbara Plank. 2022. Spectral probing. EMNLP 2022, Abu Dhabi, United Arab Emirates, December 7-11, 2022, pages 7730–7741. Association for Computational Linguistics
>
> ### Further Experiment
>
> Furthermore, we conducted an in-depth analysis of the intermediate results obtained from the Automatic Frequency Selector (AFS). From Table 7 in the manuscript, we observe that incorporating AFS consistently enhances model performance without manual adjustments to frequency components. This suggests the effectiveness of the automated Frequency Components selection process. Based on AFS's frequency component selection outcomes, we performed statistical analyses across three datasets in accordance with the spectral band distribution outlined in Table 5 of the manuscript. The table below illustrates the percentage of Frequency Components selected by AFS within different spectral bands, while "Overall" represents the percentage of selected Frequency Components across all four bands.
>
> It is evident that the results are not uniformly 100%, indicating that AFS indeed performs selection on Frequency Components, thereby adjusting information at various scales to achieve the consistent improvements highlighted in Table 7 of the manuscript. Moreover, the percentage of selected Frequency Components varies across different datasets, implying adaptive adjustments by AFS to cater to the diverse demands of distinct samples. Notably, the LOW band exhibits the highest percentage of selected Frequency Components, underscoring the significance of sentence-level information for token-level tasks (such as Structure Induction for ABSA, which can be considered a token-level task). This observation aligns with the conclusion drawn in reference [2].
>
> | Band | Rest14(%) | Laptop14(%) | Twitter(%) | Scale | DFT index |
> | --- | --- | --- | --- | --- | --- |
> | HIGH | 84.77 | 25.64 | 87.22 | Word | L/2 → L |
> | MID-HIGH | 89.82 | 28.68 | 92.61 | Phrase | L/6 → L/2 |
> | MID-LOW | 91.82 | 41.02 | 96.87 | Clause | L/14 → L/6 |
> | LOW | 99.61 | 88.08 | 99.19 | Sentence | 1 → L/14 |
> | Overall | 88.88 | 35.21 | 91.41 | - | - |
>
> The specific configuration of spectral bands in our study is detailed in Table 5 of the manuscript.
>
> ## 2： Combine multiple graphs based on the previous similarity functions and then using a fusion gate
>
> Indeed, we previously delved into inducing multiple graphs simultaneously, while also avoiding the use of off-the-shelf parsers. We achieved this integration through gating mechanisms. Typically, such an approach yields improvements. However, it's crucial to ensure that each induced graph remains distinct. Moving forward, we are committed to further exploration in this realm to refine our understanding and methodologies in inducing multiple graphs effectively.
>
> ## 3: The comparison between FLT and AFS
>
> AFS is a variant of FLT. They are based on the scale information operation of the context. For FLT, it is necessary to heuristically select Frequency Components. AFS avoids this burden and makes the method more flexible and convenient. Although the effect of carefully selecting Frequency Components under FLT may exceed AFS, the effect of AFS is still greater than Attn, and it is at the head of all possible Frequency Component selection results of FLT.
>
> ## 4: Other Concerns
>
> - **References:** Based on your suggestion, we will include all the references you pointed out in the camera-ready version.
> - **Typos Grammar Style And Presentation Improvements:** Thank you very much for pointing out our representation problems; we will fix all of them in the camera-ready version.

---

### Official Review · Reviewer_ubU3 · 2023-08-04

**Soundness:** 3

**Excitement:**

3: Ambivalent: It has merits (e.g., it reports state-of-the-art results, the idea is nice), but there are key weaknesses (e.g., it describes incremental work), and it can significantly benefit from another round of revision. However, I won't object to accepting it if my co-reviewers champion it.

**Paper Topic And Main Contributions:**

This paper introduces an ABSA parsing model that generates a latent structure. Specifically, the model employs a pre-trained language model (RoBERTa) to generate language representations. These representations are then filtered using frequency filters. Next, a GSL module is utilized to induce latent structure from the filtered representations. The paper thoroughly evaluates the proposed model through extensive experiments on three public benchmarks for ABSA.





**Reasons To Accept:**

The proposed model demonstrates strong theoretical foundations.

The experimental results provide evidence of the model's effectiveness.

**Reasons To Reject:**

1. The writing requires polishing. It's unclear whats the input and output of the proposed model. From section 4, it seems the model's output is the sentiment label, but the model description in Section 3 doesn't mention anything about classification.

2. The citation format is incorrect. For example, in line 285, it should be written as "inspired by Tamkin et al., (2020)," instead of "inspired by (Tamkin et al., 2020)."

3. Figure 1 is not clear as it lacks input/output arrows to illustrate how the data is transformed. The role and purpose of the prediction head in this architecture are not well-defined, making it difficult to understand its function within the model. More detailed explanations and visual annotations would improve the clarity and comprehension of the figure.

**Reproducibility:**

3: Could reproduce the results with some difficulty. The settings of parameters are underspecified or subjectively determined; the training/evaluation data are not widely available.

**Reviewer Confidence:**

3: Pretty sure, but there's a chance I missed something. Although I have a good feel for this area in general, I did not carefully check the paper's details, e.g., the math, experimental design, or novelty.

---

> ### Author Rebuttal · Authors · 2023-08-29
>
> Dear reviewer, thank you so much for your detailed comments and affirmation of our approach.
>
> ## 1: Model Description
>
> We will carefully polish the camera-ready version and describe the input and output of the model in detail in the camera-ready version. This task (ABSA) is a sentiment classification task for aspect words in a sentence. Since the paper describes the focus on structure induction and Frequency Filter (FLT), there are fewer descriptions of the input and output of the model and the definition of the problem. We will The description of this part will be added in the camera-ready version.
>
> ## 2: Citation Format
>
> We will modify the citation format issue in the camera-ready version, and if we check for the same similar issues, we will also modify them.
>
> ## 3: Figure 1 in our manuscript
>
> The input of the model is only the binary group (sentence, aspect word), and the emotional polarity label (Positive, Negative, and Neutral) of the aspect word is used as the classification label. The output of the model prediction head is the logits for classification. The cross-entropy loss will be calculated with the aspect word sentiment polarity label so that the whole model can perform gradient backpropagation according to the loss. Regarding the problem of Figure 1, we will improve it in the camera-ready version to make the diagram more clearly represent data transformation and input and output.

---

### Official Review · Reviewer_n3xo · 2023-08-04

**Soundness:** 3

**Excitement:**

3: Ambivalent: It has merits (e.g., it reports state-of-the-art results, the idea is nice), but there are key weaknesses (e.g., it describes incremental work), and it can significantly benefit from another round of revision. However, I won't object to accepting it if my co-reviewers champion it.

**Missing References:**

1. Binxuan Huang and Kathleen M Carley. SyntaxAware Aspect Level Sentiment Classification with Graph Attention Networks. EMNLP-2019
2. Amir Pouran Ben Veyseh, Nasim Nouri, Franck Dernoncourt, Quan Hung Tran, Dejing Dou, Thien Huu Nguyen. Improving Aspect-based Sentiment Analysis with Gated Graph Convolutional Networks and Syntax-based Regulation. EMNLP-2020: findings
3. Yuanhe Tian, Guimin Chen, Yan Song. Aspect-based Sentiment Analysis with Type-aware Graph Convolutional Networks and Layer Ensemble. NAACL-2021

**Paper Topic And Main Contributions:**

The paper proposes to improve aspect-based sentiment analysis (ABSA) through the induction of structure information, where graph structure learning (GSL) module that consists of a graph learner and Graph Neural Networks (GNNs) is proposed. The paper incorporated Frequency Filters to produce filtered language representations and feed these into the GSL module to generate latent structures. Experiments on three ABSA benchmark datasets demonstrated that this approach reduced the aspect-sentiment distance (AsD), leading to good performance.

**Reasons To Accept:**

1. The proposed approach achieved good performance on English benchmark datasets.
2. The paper is easy to follow.


**Reasons To Reject:**

1. Many techniques (e.g., attentions, graph neural networks, discrete Fourier transform) used in the paper are not new.
2. It is not clear how the hyper-parameters are tuned. It seems the results reported in Table 2 and 3 comes from a single run (the authors do not report the average and standard deviation of the results). Given the test sets of the benchmark datasets are small, it is possible that the model overfits the test set.

**Reproducibility:**

3: Could reproduce the results with some difficulty. The settings of parameters are underspecified or subjectively determined; the training/evaluation data are not widely available.

**Reviewer Confidence:**

4: Quite sure. I tried to check the important points carefully. It's unlikely, though conceivable, that I missed something that should affect my ratings.

---

> ### Author Rebuttal · Authors · 2023-08-29
>
> ## 1: Hyperparameter in FLT
>
> For different manually selected frequency components in FLT, please refer to Table 8 in the appendix. For your convenience, we have also presented the results from Table 8 below, with bold values indicating the frequency selections that demonstrate improvements of FLT relative to Attn.
>
>   (Detailed results of the band impact based on RoBERTa(base) for heuristic frequency selection. For real
> sequence, the spectrum obtained by the Discrete Fourier Transform is symmetrical, so we only take half of the spectral
> bands for analysis. Negative values indicate that the frequency is selected from the high-frequency band, and
> positive values mean that the frequency is selected from the low-frequency band. Additionally, x → y means that
> the frequency selection is between the two values (x and y))
>
> | Filter | Bands | Rest14 Accuracy | Rest14 Macro-F1 | Laptop14 Accuracy | Laptop14 Macro-F1 | Twitter Accuracy | Twitter Macro-F1 |
> |--------|-------|-----------------|-----------------|-------------------|-------------------|------------------|------------------|
> | HIGH   | -1    | 87.32           | 80.76           | **84.48**             | **81.54**             | 75.43            | **74.88**            |
> |        | -2    | 87.32           | 80.79           | **84.17**             | **81.13**             | **75.72**            | **74.45**            |
> |        | -3    | 87.23           | 81.56           | **83.86**             | **81.20**             | **76.01**            | **74.34**            |
> |        | -4    | 86.88           | 80.44           | **84.01**             | **81.34**             | 75.43            | **74.78**            |
> |        | -5    | **87.77**           | 81.62           | 83.54             | 80.53             | **76.30**            | **75.00**            |
> |        | -6    | **87.77**           | 81.71           | 82.76             | 79.93             | 75.58            | **74.34**            |
> |        | -8    | **87.77**           | 80.74           | **84.80**             | **82.27**             | **76.16**            | **75.52**            |
> |        | -10   | 87.05           | 80.79           | **83.86**             | **81.37**             | 75.58            | **74.41**            |
> |        | -12   | **87.77**           | 80.74           | **84.48**             | **81.38**             | **75.87**            | **74.45**            |
> |        | -14   | **87.75**           | **81.86**           | **84.80**             | **82.21**             | 75.43            | **74.69**            |
> |        | -16   | **88.57**       | **82.95**           | 84.32             | **81.87**             | **76.45**            | **75.46**            |
> |        | -18   | 86.43           | 79.26           | 83.54             | **80.54**             | 75.43            | **74.08**            |
> |        | -20   | **88.13**           | **82.33**           | **84.01**             | **81.06**             | **76.01**            | **75.23**            |
> |        | -22   | **88.57**       | **83.27**       | **84.48**             | **81.82**             | 75.58            | **74.91**            |
> |        | -24   | 87.14           | 80.63           | **84.17**             | **81.65**             | **76.30**            | **75.18**            |
> |        | -26   | 87.50           | 80.85           | **84.64**             | **82.04**             | **76.01**            | **74.46**            |
> |--------|---------|-----------------|-----------------|-------------------|-------------------|------------------|------------------|
> | MID-HIGH | 8 \to 10 | **88.21**    | **82.41**           | **84.48**             | **81.90**             | 74.57            | **74.19**            |
> |          | 8 \to 11 | **87.86**         | 81.69           | **85.42**         | **83.01**         | 75.29            | **74.59**            |
> |          | 8 \to 12 | 87.50         | 80.66           | 83.39             | 80.49             | 75.29            | **74.68**            |
> |          | 8 \to 13 | 87.23         | 80.13           | **83.86**             | **81.06**             | **76.88**           | **75.70**            |
> |          | 8 \to 14 | 86.88         | 80.75           | **84.48**             | **81.70**             | **75.72**            | **74.90**            |
> |          | 8 \to 16 | **87.95**         | 81.69           | 83.70             | 80.92             | **77.02**        | **75.84**        |
> |          | 8 \to 18 | 87.50         | **82.16**           | **85.27**             | **82.67**             | **75.72**            | **74.48**            |
> |          | 8 \to 20 | **88.48**         | **83.32**           | 83.70             | 80.81             | 75.14            | 73.63            |
> |          | 8 \to 22 | 87.05         | 79.81           | 83.54             | 80.50             | **76.45**            | **75.16**            |
> |          | 8 \to 24 | 86.88         | 80.53           | **84.33**             | **81.65**             | 75.00            | 73.62            |
> |--------|---------|-----------------|-----------------|-------------------|-------------------|------------------|------------------|
> | HIGH   | 4 \to 5 | 86.96           | 80.50           | **84.01**             | **81.14**             | **76.45**            | **75.50**            |
> |        | 4 \to 6 | 87.14           | 80.40           | 83.70             | **81.05**             | **76.59**            | **75.61**            |
> |        | 4 \to 7 | 87.14           | 81.71           | **84.33**             | **82.10**             | **77.02**        | **75.64**            |
> |        | 4 \to 8 | **87.68**           | **81.99**           | 82.92             | 79.72             | **76.87**            | **75.82**            |
> |  LOW   | 1       | 87.41           | 81.27           | 83.39             | 80.44             | **76.16**            | **75.03**            |
> |        | 2       | **87.86**           | 81.06           | 83.39             | **80.55**             | **76.15**            | **75.16**            |
> |        | 3       | 87.23           | 80.51           | 83.70             | **80.80**             | **76.45**            | **74.90**            |
> |        | 4       | 86.96           | 80.14           | **84.01**             | **81.64**             | **75.87**            | **74.65**            |
> | Attn.  | -       | 87.59           | 81.72           | 83.86             | 80.53             | 75.72            | 73.92            |
>
>
> ## 2: Benchmark Dataset and Results
>
> Based on prior related research and our experimental findings, ensuring a consistent random seed guarantees reproducibility, making each run of the model yield identical results. Furthermore, the datasets we employed are well-established and commonly used in previous related works. Comparing the results against all baselines was conducted under uniform conditions, employing the same datasets and experimental settings. Consequently, our experimental outcomes are both reliable and stable.
>
> ## 3: References
>
> Based on your suggestion, we will include all the references you pointed out in the camera-ready version.

---

### Meta-Review · Area_Chair_wvqc · 2023-09-18

**Recommendation:** 4

**Metareview:**

The paper introduces an approach to enhance aspect-based sentiment analysis (ABSA) by incorporating graph structure learning (GSL) techniques. The proposed method utilizes a Graph Neural Network (GNN) within a GSL module to induce latent structures from filtered language representations, generated using Frequency Filters. These latent structures aim to improve the accuracy of ABSA. The paper evaluates the approach on three benchmark ABSA datasets and reports favorable results.
Key strengths of the paper include its solid theoretical foundation and promising experimental results. However, reviewers raised concerns about the clarity of the paper's explanations (lack of clarity regarding hyper-parameter tuning), the novelty of the techniques used, and issues with presentation (need for writing revisions to improve clarity), which could be improved for the camera-ready version if accepted.

---

### Decision · Program_Chairs · 2023-10-07

**Decision:**

Accept-Findings

**Comment:**

The paper introduces an approach to enhance aspect-based sentiment analysis (ABSA) by incorporating graph structure learning (GSL) techniques. The proposed method utilizes a Graph Neural Network (GNN) within a GSL module to induce latent structures from filtered language representations, generated using Frequency Filters. These latent structures aim to improve the accuracy of ABSA. The paper evaluates the approach on three benchmark ABSA datasets and reports favorable results.
Key strengths of the paper include its solid theoretical foundation and promising experimental results. However, reviewers raised concerns about the clarity of the paper's explanations (lack of clarity regarding hyper-parameter tuning), the novelty of the techniques used, and issues with presentation (need for writing revisions to improve clarity), which could be improved for the camera-ready version if accepted.